# Congestive Hepatopathy

**DOI:** 10.3390/ijms21249420

**Published:** 2020-12-10

**Authors:** José Ignacio Fortea, Ángela Puente, Antonio Cuadrado, Patricia Huelin, Raúl Pellón, Francisco José González Sánchez, Marta Mayorga, María Luisa Cagigal, Inés García Carrera, Marina Cobreros, Javier Crespo, Emilio Fábrega

**Affiliations:** 1Gastroenterology and Hepatology Department, University Hospital Marqués de Valdecilla, 39008 Santander, Spain; angelapuente@hotmail.com (Á.P.); antonio.cuadrado@scsalud.es (A.C.); patricia.huelin@scsalud.es (P.H.); ines.garciac@scsalud.es (I.G.C.); marina.cobrerosdel@scsalud.es (M.C.); javiercrespo1991@gmail.com (J.C.); emilio.fabrega@scsalud.es (E.F.); 2Group of Clinical and Translational Research in Digestive Diseases, Health Research Institute Marqués de Valdecilla (IDIVAL), 39011 Santander, Spain; 3Biomedical Research Networking Center in Hepatic and Digestive Diseases (CIBERehd), 28029 Madrid, Spain; 4Radiology Department, University Hospital Marqués de Valdecilla, 39008 Santander, Spain; raul.pellon@scsalud.es (R.P.); franciscojose.gonzalezs@scsalud.es (F.J.G.S.); 5Pathological Anatomy Service, University Hospital Marqués de Valdecilla, 39008 Santander, Spain; martam.mayorga@scsalud.es (M.M.); mluisa.cagigal@scsalud.es (M.L.C.)

**Keywords:** cirrhosis, portal hypertension, heart failure, heart transplantation

## Abstract

Liver disease resulting from heart failure (HF) has generally been referred as “cardiac hepatopathy”. One of its main forms is congestive hepatopathy (CH), which results from passive venous congestion in the setting of chronic right-sided HF. The current spectrum of CH differs from earlier reports with HF, due to ischemic cardiomyopathy and congenital heart disease having surpassed rheumatic valvular disease. The chronic passive congestion leads to sinusoidal hypertension, centrilobular fibrosis, and ultimately, cirrhosis (“cardiac cirrhosis”) and hepatocellular carcinoma after several decades of ongoing injury. Contrary to primary liver diseases, in CH, inflammation seems to play no role in the progression of liver fibrosis, bridging fibrosis occurs between central veins to produce a “reversed lobulation” pattern and the performance of non-invasive diagnostic tests of liver fibrosis is poor. Although the clinical picture and prognosis is usually dominated by the underlying heart condition, the improved long-term survival of cardiac patients due to advances in medical and surgical treatments are responsible for the increased number of liver complications in this setting. Eventually, liver disease could become as clinically relevant as cardiac disease and further complicate its management.

## 1. Introduction

The interactions between the heart and the liver have been known for a long time. In recent years, however, these cardio-hepatic interactions have gained greater interest, which has led to a better understanding of their pathophysiology. They are usually classified into three groups, according to the role of each organ as culprit or victim of the other [1,2]: (1) liver disease resulting from heart disease; (2) heart disease resulting from liver disease (e.g., cirrhotic cardiomyopathy); (3) systemic diseases that affect both the heart and liver (e.g., systemic amyloidosis). The former group has generally been referred as “cardiac hepatopathy”, although there is still no consensus on terminology [3,4]. The two main forms of cardiac hepatopathy are acute cardiogenic liver injury (ACLI) (also referred as hypoxic hepatitis) and congestive hepatopathy (CH). Both conditions often coexist and potentiate the deleterious effects of each other on the liver [3,4,5].

This article seeks to make a comprehensive review of the pathophysiology, clinical features, diagnosis and treatment of congestive hepatopathy (CH). This clinical entity is characterized by congestion of the liver parenchyma induced by impaired hepatic venous outflow secondary to right-sided heart failure (HF). As will be discussed, despite the progress achieved since the earlier studies from the beginning of the 20th century that provided the first data on its structural and functional changes [6,7], there are still important gaps in the diagnosis and management of this form of liver disease [4,8].

## 2. Epidemiology

Any cause of right-sided HF (e.g., constrictive pericarditis, mitral stenosis, severe tricuspid regurgitation, congenital heart disease, or end-stage cardiomyopathies) can lead to CH [9,10]. The widespread use of heart transplantation (HT) and major advances in medical and surgical treatments have significantly changed the profile of patients harboring CH. Thus, compared to earlier reports, cardiac cirrhosis due to non-congenital HF is declining, ischemic cardiomyopathy is now the leading cause of HF having surpassed rheumatic HF, and CH following Fontan surgery is on the rise [1,2,4,11]. 

The latter surgery is used to treat several complex congenital heart diseases with a functional single ventricle (e.g., tricuspid or mitral atresia and hypoplastic left or right heart syndrome). It is usually performed in children 2 to 5 years in whom a superior cavopulmonary connection has been previously performed through the Glenn procedure. The Fontan technique then creates a total cavopulmonary connection by implanting a surgical shunt to divert blood from the inferior and superior vena cava to the pulmonary arteries, which passively carry the blood to the single ventricular chamber. This bypass leads to chronic hepatic venous congestion secondary to high-pressure nonpulsatile flow in the inferior vena cava. The lack of a subpulmonary ventricle also leads to diminished cardiac preload for the systemic ventricle, resulting in chronically low cardiac output. These hemodynamic changes together with the characteristic mild low arterial blood oxygen saturation are responsible for the damage that can affect virtually all organs. As far as the liver is concerned, the functional and structural alterations that systematically develop after this surgery are referred as Fontan-associated liver disease. Its natural history is poorly understood, and we are presently unable to predict and correctly identify the patients that will develop clinically significant advanced liver disease [12,13,14].

In non-congenital HF, there are no reliable data on the prevalence of CH, with even fewer solid data concerning the stage of liver disease. This is mainly due to the limited validated techniques available to diagnose and, specially, stage the disease [15]. Studies using liver blood tests have described prevalence figures of CH ranging from 15 to 80%, depending on the severity of HF [16,17,18,19,20,21,22]. However, liver blood tests neither accurately diagnose CH nor reflect the stage of liver disease [15].

## 3. Pathophysiology

The liver is a highly vascular organ that receives up to 25% of the total cardiac output from a dual blood supply. The hepatic artery delivers well-oxygenated blood and comprises approximately 25% of total hepatic blood flow, whereas the remaining 75% is deoxygenated blood supplied by the portal vein [5]. Its robust vascular mechanisms of defense make the liver resilient to ischemic damage [1]. The hepatic artery buffer response is one of such mechanisms that may be capable of compensating for up to a 60% decrease in portal flow. It refers to the compensatory up-regulation of hepatic arterial flow instigated by any decrease in portal flow. [1,5,10]. The signaling pathway for this response is local, with the reduction of portal flow resulting in an increase in concentration of the vasodilator adenosine [23]. In contrast, the portal vein does not have the ability to autoregulate its flow, and is dependent on cardiac output and the gradient between portal and hepatic venous pressures [5,10]. The high permeability of sinusoids enabling oxygen extraction to levels near 90% represents a second mechanism of defense against hypoxia and prevents any change in liver oxygen consumption, despite decreases in liver blood flow up to half its normal [5,24,25]. 

This unique resilience to ischemic damage contrasts with the paucity of protective mechanisms against congestion. These mainly rely on the highly connected sinusoidal network to relieve the pressure surge that hits the sinusoidal bed without attenuation, since the hepatic veins lack valves [4]. The resulting congestion produces liver damage through several pathogenic mechanisms: (1) shear stress promotes fibrogenesis and sinusoidal ischemia by the activation of hepatic stellate cells and by a decrease in nitric oxide production from endothelial cells [15,26]; (2) decreased portal and arterial inflow aggravates hepatic ischemia. The former is due to a reduced hepatic venous pressure gradient as a result of the transmission of the elevated central venous pressure to the sinusoidal network, while the latter can also be compromised in patients with left-sided HF [10,15]; (3) impairment of oxygen and nutrients diffusion due to the accumulation of exudate into the space of Disse further promotes fibrogenesis [10]; (4) sinusoidal stasis and congestion promote sinusoidal thrombosis, which, in turn, contribute to liver fibrosis by causing parenchymal extinction, and by activating hepatic stellate cells via protease-activated receptors [27,28]. The former refers to a hypothesis based on retrospective observations of ex-vivo human liver specimens of patients with CH. In this autopsy study, Wansless et al. demonstrated sinusoidal thrombi confined to areas of fibrosis, thereby suggesting that intrahepatic thrombosis is involved in liver fibrosis progression [29]. This author has recently updated his “vascular hypothesis” for the pathogenesis of cirrhosis. He postulates that cirrhosis of any etiology is the morphologic result of parenchymal extinction. The definition of the latter has been modified to recognize the importance of sinusoidal destruction in driving fibrogenesis. It is now defined as a region with focal loss of contiguous hepatocytes and adjacent microvascular structures. The new model incorporates the concept of a “congestive escalator”, whereby the initial damage is usually at the level of sinusoidal endothelial cells and progresses to parenchymal extinction by a sequence of events that involve vascular leak, transudation into vein walls and interstitium, ischemia, and hyperemia. This microvascular injury leads to the extension of venous obstruction to larger vessels perpetuating and aggravating the congestive injury [30]. A recent experimental study provided evidence of the mechanistic link between CH and liver fibrosis through some of these mechanisms [26]. 

It must be highlighted that contrary to primary liver diseases, in CH inflammation seems to play no role in the progression of liver fibrosis. Indeed, several studies of patients with Fontan circulation demonstrated minimal inflammatory changes in liver biopsy specimens, despite accentuated hepatic fibrosis [31,32,33]. All these findings settle the rational basis for testing anticoagulant drugs in patients with CH, but so far, no clinical trial has addressed this issue. In comparison, research in this area in primary liver cirrhosis is more advanced. Hence, several experimental studies have shown that anticoagulant therapy improves liver fibrosis and reduces portal hypertension [34,35,36,37,38,39,40,41,42,43,44,45], and a clinical trial demonstrated that anticoagulation led to a reduction in portal thrombosis and other complications of liver disease, and to increased survival [46]. New clinical trials (CIRROXABAN, NCT02643212) are on their way in order to confirm these preliminary results [47]. 

The role of cardiokines in the pathophysiology of CH remains uncertain. These proteins are secreted by the heart for inter-organ and inter-cellular communication. To date, more than 16 cardiokines have been identified, with natriuretic peptides being the most well-studied (atrial natriuretic factor and B-type natriuretic peptides). Increasing evidence suggests that cardiokines are involved in the metabolic crosstalk between myocardial inflammation in HF and peripheral tissue damage in some organs (adipose, tissue, skeletal muscle, spleen, and kidney), but the direct mechanisms linking heart and liver disease remain not fully characterized. Experimental evidence suggests that cardiokines enhance lipid uptake and β-oxidation, and regulate other genes involved in fatty acids utilization [48,49,50,51,52]. In the setting of CH, the impact of these metabolic alterations in liver disease progression and body wasting deserve further study. 

Finally, the pre-existing hepatic congestion predisposes the liver to ACLI [5,53]. This other form of cardiac hepatopathy is the result of several mechanisms that often concur with uneven preeminence, depending on the underlying condition: passive congestion, reduced hepatic blood, total body hypoxemia, inability to utilize oxygen, and ischemia/reperfusion injury. The most frequent cause leading to ACLI is HF (39–78%), followed by septic toxic shock (15–30%) and respiratory failure (15%) [5,53]. As shown by Fuhrmann et al., these causes often coexist, as it occurred in 74% of their study population [54]. In the setting of HF, ACLI is believed to reflect the extreme of a spectrum of liver injury that begins with passive hepatic congestion since the vast majority of patients have markedly elevated cardiac filling pressures [55,56,57,58,59]. Thus, several studies have shown how, despite similar hemodynamic derangements, only those with a pre-existing congestive liver developed ACLI [57,60,61]. This crucial role of passive congestion of the liver justifies the rare occurrence of ACLI in hemorrhagic or hypovolemic shock and the frequent presence of CH in ACLI due to respiratory failure or septic shock [5]. 

## 4. Clinical Presentation and Diagnosis

CH may be asymptomatic for a long time and in these patients the only clue to suspect its presence might be through abnormalities in liver tests [10]. When symptomatic, the digestive symptoms are usually masked by those related to right-sided HF [8]. The stretching of the liver capsule due to hepatic congestion is responsible for some digestive symptoms such as dull right upper quadrant pain and nausea. Other symptoms include anorexia, early satiety, and malaise. Of note, all of them may occur in the absence of overt ascites or lower extremity edema [1]. Physical examination may often show hepatomegaly and signs of HF, including hepatojugular reflux, peripheral edema and ascites. The latter is a frequent finding and does not necessary denote that cardiac cirrhosis has already developed. It is most commonly due to elevated right-sided cardiac pressure hitting the sinusoidal network. Indeed, in a series of 83 patients with CH of whom only one had cardiac cirrhosis, up to 57% had ascites and its presence had no relation to the extent of liver fibrosis [11]. Classical complications of cirrhosis (e.g., hepatic encephalopathy or hepatocarcinoma) occur in late stages of cardiac cirrhosis, and may eventually become as clinically important as the cardiac disease and further complicate its management [15]. Thanks to advances in medical and surgical treatments, this clinical scenario is becoming more frequent, since the longer survival of patients with cardiac cirrhosis increases the likelihood of progressing to decompensated cirrhosis or developing a hepatocarcinoma [1]. 

When facing a patient with new-onset ascites it might be cumbersome to differentiate cardiac ascites from cirrhotic ascites as in both conditions the serum-ascites albumin gradient is ≥1.1 g/dL as a result of hepatic sinusoidal hypertension [62]. However, cardiac ascites has higher protein levels (>2.5 g/dL), due to preserved liver synthetic function and the absence of capillarization of the liver sinusoidal endothelial cells [1,10,63]. The latter is characterized by a decrease in the permeability of these cells, due to the loss of fenestrae and the development of a basement membrane. These morphological changes prevent the passage of proteins to the space of Disse, and from here to the peritoneal fluid, thus, explaining the lower concentrations of proteins in ascites due to cirrhosis [64]. Other less reliable findings in cardiac ascites are higher LDH levels and higher red blood cell counts, due to leaking of red blood cells into the ascites via lymph tissue, with resulting lysis [63]. More recently, measurement of serum B-type natriuretic peptide (BNP) or of its inactive pro-hormone (N-terminal-proBNP) in serum and ascites has been suggested as an aid tool in uncertain cases. Thus, Sheer et al. reported that both serum and ascites NT-proBNP levels had high sensitivity and specificity in predicting HF as the cause of ascites [65]. Similarly, Farias et al. found serum BNP to be superior to the total ascitic fluid protein concentration with regard to discriminating cardiac ascites from cirrhotic ascites. A serum BNP cutoff of >364 pg/mL had 98% sensitivity, 99% specificity, 99% diagnostic accuracy, and a positive likelihood ratio of 168.1 for the diagnosis of cardiac ascites. Conversely, a serum BNP cutoff of ≤ 182 pg/mL was excellent for ruling out ascites due to HF [62]. 

The differentiation of cardiac cirrhotic ascites from cardiac ascites without cirrhosis is especially challenging, and might require invasive diagnostic tests, such as liver biopsy and hepatic venous pressure gradient (HVPG). The absence of neither stigmata of chronic liver disease (e.g., spider angiomata) nor imaging findings suggestive of portal hypertension (e.g., splenomegaly or porto-systemic collaterals), supports the diagnosis of cardiac ascites without cirrhosis [1,9]. The low prevalence of gastroesophageal varices in this population can be explained by the fact that varices represent collateral vessels from the high-pressure portal system to the low-pressure systemic circulation, and in CH without cirrhosis no pressure gradient exists because pressure remains high along the entire path of venous return to the right atrium [11]. 

In addition to the presence of right-sided HF (or other cause of high central pressures) and the aforementioned clinical findings, the diagnosis of CH should be further supported on compatible results of diagnostic tools and exclusion of other possible causes of liver disease [9,11].

## 5. Biochemical Profile

Laboratory examinations may remain within the normal ranges and are of little value for both diagnosing and staging the stage of liver disease. The most common laboratory abnormality is a mild hyperbilirubinemia (rarely exceeding 3 mg/dL) with a predominantly unconjugated fraction. Elevation of other serum cholestasis markers (alkaline phosphatase and gamma-glutamyl transferase) often coexist [1]. The degree of cholestasis is related to the severity of both the elevation of right atrial pressure and tricuspid regurgitation [20,66]. These data suggest that elevated right-sided filling pressures may contribute more to the elevation of liver enzymes than reduced cardiac output [8]. The mechanism of cholestasis in this setting is thought to be due to the compression of the bile canaliculi and small ductules by centrally congested sinusoids [67]. Other laboratory findings include mild elevations of serum aminotransferases to two to three times the upper limit of normal and mild hypoalbuminemia. The latter may also be secondary to malnutrition or protein-losing enteropathy [10]. As liver disease progresses, liver function tests (i.e., bilirubin, international normalized ratio, and albumin) may continue to worsen. 

As already discussed, CH predisposes the liver to ACLI in the face of different settings. Its biochemical profile is characterized by a substantial and rapid increase in aminotransferases and lactate dehydrogenase (LDH) levels to 10 to 20 times the upper limit of normal, usually 1 to 3 days after hemodynamic deterioration. Importantly, the latter is far from being a constant feature, as a shock state is only observed in half of the cases. This is probably due to the fact that short periods of hypotension (i.e., 15–20 min), which can be easily unrecognized, are sufficient to provoke ACLI [57]. Thus, the diagnosis of ACLI cannot be rejected because of absence of shock, and, in case of uncertainty, a cardiac evaluation is warranted [4,5]. Once hemodynamic stability is restored, these laboratory abnormalities generally return to normal within 7 to 10 days [1,68]. A progressive increase in bilirubin is usually seen but is seldom severe [1,5,53]. The higher values reported by recent series may be explained by the inclusion of more patients with septic shock. Nonetheless, the mean bilirubin value in these studies was lower than 6 mg/dl [54,69]. Higher values may suggest progression to acute liver failure [4]. Unlike in children where hypoglycemia has been regarded as a distinct feature of ACLI, in adults both hypoglycemia and hyperglycemia have been reported [5,53]. Although no analytical alteration is pathognomonic of ACLI, there are some findings that suggests its diagnosis [5]: (1) an alanine aminotransferase (ALT)-to-LDH ratio <1.5 is of great help in the differential diagnosis, as it is rarely seen in other etiologies of hepatitis [70]; (2) the aspartate aminotransferase (AST) generally peaks earlier and higher than ALT [68]. The rationale behind this finding lies in the concentration of aminotransferases throughout the hepatic acinus. ALT reaches the highest concentration at the level of periportal hepatocytes (Rappaport liver zone 1) and the lowest concentration at the level of pericentral hepatocytes (Rappaport liver zone 3), while AST maintains a stable concentration throughout the entire acinus. Hence, after the hypoxic insult the initial concentrations of AST are higher than those of ALT, since the lower oxygen concentration of pericentral hepatocytes make them more susceptible to hypoxic damage [71]. Once the cause of liver damage is resolved, the concentration of ALT exceeds that of AST in subsequent days, due to its longer half-life (47 ± 10 h versus 17 ± 5 h, respectively) [72]. Aboelsoud et al. [68] universally observed this pattern, but it was only described in 75% of the cases in Henrion’s study [58]. The rapid decline and reversal of the AST-ALT ratio may explain these differences and, therefore, an ALT higher than AST should not discard ACLI; (3) an early and sharp deterioration in prothrombin activity and renal function also supports ACLI. Such abnormalities are unusual at presentation in patients with viral or drug-induced hepatitis, unless acute liver failure is already established [5]. Figure 1 shows a typical biochemical profile of ACLI in a patient treated in our hospital.

In accordance with the above, diagnosis of ACLI is usually made when the following criteria are met [53,58,59]: (1) an appropriate clinical setting of cardiac, respiratory or circulatory failure; (2) severe increase in aminotransferase levels; (3) exclusion of other causes of acute liver damage. It should be noted that the differential diagnosis for severe elevations of transaminases is relatively limited and includes, besides ACLI, acute viral hepatitis, toxin- or drug-induced liver injury, autoimmune hepatitis, Wilson’s disease, acute bile duct obstruction and acute Budd-Chiari syndrome [72].

## 6. Imaging Tests

Imaging tests help both to support the diagnosis of CH and to identify complications. It is of great help to specify the clinical suspicion of CH in the radiological request since many of the findings are elusive. Importantly, none of them is specific to CH and, therefore, the diagnosis should also be supported by other extrahepatic findings such as cardiomegaly, hypertrophy of the right atrium and ventricle, thickening and calcification of the pericardium, pericardial effusion, or pleural effusion [73] (Figure 2A). 

Abdominal ultrasound is typically the first imaging modality used to evaluate patients with suspected liver disease. It provides important information regarding the morphology and vascularization of the liver as the Doppler mode can evaluate the direction and speed of the hepatic blood flow. Characteristic ultrasound findings include hepatomegaly, an irregular and nodular liver, dilation of inferior vena cava and hepatic veins with absence or attenuation of the normal variation of their diameter with respiratory movements, loss of normal triphasic hepatic venous waveform (under physiological conditions the hepatic veins present a predominantly anterograde flow with a triphasic wave pattern, in which four waves can be identified—“a”, “S”, “v” and “D”, each corresponding to a different phase of the cardiac cycle), and increase in the portal vein pulsatility index and in hepatic arterial resistance [73,74] (Figure 2B–D). Of note, the appearance of a nodular or heterogeneous liver on standard imaging is not sufficient to diagnosis cirrhosis in CH [15].

Computed tomography and magnetic resonance imaging allow a better morphological characterization of the liver and also identify abnormal kinetics of intravenous contrast enhancement. These include delayed bolus arrival to the liver suggesting slow systemic circulation, diffusion of extracellular contrast media into the periportal lymphatic space in the delayed phase, early enhancement of the inferior vena cava and hepatic veins as a result of the reflux of the contrast from the atrium and a predominantly peripheral heterogeneous pattern of hepatic enhancement, due to stagnant blood flow [74] (Figure 3A,B). The latter is best evaluated in the portovenous phase. Both techniques improve the identification of the frequent hypervascular nodules that develop in this setting. Indeed, CH may lead to the generation of benign regenerative nodules or focal nodular hyperplasia (FNH)-like lesions, and hepatocarcinoma. The former are referred to as “FNH-like” despite having characteristic pathological findings of FNH due to the presence of abnormal background liver parenchyma. Although they most commonly demonstrate typical imaging findings (i.e.**,** well-circumscribed, homogeneous nodule with late arterial hyperenhancement which fades to isointensity/isoattenuation on delayed phase imaging), they sometimes have a washout appearance that could be mistaken for hepatocarcinoma due to abnormally increased background parenchymal enhancement in the delayed phase [74] (Figure 4). Indeed, distinguishing hepatocarcinoma from these atypical imaging represents an unmet need and biopsy is frequently required for accurate diagnosis. Radiological findings that support the diagnosis of hepatocarcinoma include the following: significant change in appearance of a nodule, venous invasion, a heterogeneous-appearing mass, and elevated alpha-fetoprotein [15,74]. There are currently no guidelines for screening for hepatocarcinoma in CH. In post-Fontan patients some experts recommend to begin screening at 15–20 years after the operation [15], while the newly released guidelines from the American Heart Association recommend a much more comprehensive surveillance (Table 1) [75]. In patients with CH due to other conditions, it seems reasonable to perform bi-annual screening once cardiac cirrhosis is established. 

## 7. Histology

The congestive liver explant has been characterized as a “nutmeg liver”, due to the presence of dark centrilobular zones that reflect sinusoidal congestion alternating with pale periportal zones with normal or fatty liver tissue [74] (Figure 5A). Characteristic histological findings include sinusoidal dilatation and congestion, hepatocyte atrophy most prominent in zone 3, extravasation of red blood cells into the space of Disse, regenerative hyperplasia emerging from periportal regions, and centrilobular fibrosis (Figure 5B,C) [67]. The degree of sinusoidal dilatation is positively correlated with the degree of elevation of right atrial pressure. As liver disease progresses, bridging fibrosis typically extends between central veins to produce a pattern that has been name “reversed lobulation”, since it contrasts to the typical fibrosis pattern found in most primary liver diseases, where bridging fibrosis occurs between portal triads (i.e., zone 1) [1]. As far as the correlation between fibrosis extension and systemic hemodynamic parameters is concerned, there are discordant results with most studies finding no correlation [11,19,76,77,78,79]. Of note, any disorder causing hepatic venous outflow obstruction (e.g., sinusoidal obstruction syndrome or Budd–Chiari syndrome) leads to similar histological findings (Figure 6) [80].

Traditional scores to evaluate the severity of fibrosis, such as METAVIR, may not be accurate enough in the setting of CH, especially in intermediate stages of fibrosis. This is due to the fact that they do not adequately reflect the reversed lobulation pattern of fibrosis observed in CH. Dai et al. recently introduced a four-tiered system for histologically scoring liver fibrosis in patients with CH, the Congestive Hepatic Fibrosis Score [77]. Although this scoring system has been increasingly utilized in recent clinical outcome studies and assessed for reproducibility among pathologists, it has not yet been widely applied in the clinical setting [78]. It must be highlighted that the distribution of fibrosis throughout the liver is extremely heterogeneous in patients with CH [76,81] and it may be explained by the fibrogenic effects of intrahepatic thrombosis caused by static blood flow [29]. This variability raises concern about sampling error and about the role of liver biopsy as the gold standard tool for fibrosis assessment. Moreover, liver biopsies may not predict post-HT outcomes. In a retrospective study, Louie et al. found that the presence of bridging fibrosis was not significantly associated with post-operative survival or post-operative liver failure, based on which they concluded that patients with bridging fibrosis may still be considered viable candidates for isolated HT [81]. Similar results were described by Dhall et al**.** [76]. Regardless of these limitations, liver biopsy still plays an important role in the assessment of the stage of liver disease, in ruling out hepatocarcinoma and alternative etiologies of liver disease and in determining candidacy for isolated HT or combined heart-liver transplantation (CHLT). Its findings, however, should be correlated with the clinical presentation and results of other diagnostic tools [15,76]. 

In patients with concomitant ACLI, the histological sections will also show features of coagulative necrosis of centrilobular hepatocytes without significant inflammation (Figure 7A–C). In biopsies delayed several days, however, there may be neutrophils infiltrating the affected regions [67]. In rare cases, necrosis occurs predominantly in the middle zone [82,83,84]. Henrion et al. postulated that this atypical histological pattern could be due to an ischemia/reperfusion injury secondary to an incomplete liver reperfusion prior to death that only reached periportal and mediolobular liver cells. Hence, periportal and centrilobular cells would have survived, the former because oxygen delivery remained sufficient, and the latter because of the absence of reperfusion injury. Mediolobular hepatocytes, on the other hand, would have been destroyed due to ischemia/reperfusion injury [5].

## 8. Non-Invasive Assessment of Liver Fibrosis

Unlike patients with viral hepatitis and non-alcoholic fatty liver disease in whom non-invasive diagnostic tests of liver fibrosis have excellent predictive value for advanced fibrosis, the performance of these tests in assessing the severity of fibrosis in CH is poor [85]. A detail description of each of these tests in this setting is beyond the scope of this review and can be found elsewhere [15,86,87]. 

Among serological markers, the model for end-stage liver disease (MELD)-XI score may be the only clinical risk calculator to have some correlation with biopsy-determined fibrosis staging [88,89]. This score excludes the international normalized ratio, given the high prevalence of anticoagulation use in CH. It must be pointed out, however, that this correlation is only moderate, and other small studies have provided opposite results [33,81]. The remaining tests (i.e., standard serum markers, low platelet count, Fibrosure testing, hyaluronic acid levels, and most clinical risk calculators) are inaccurate at staging liver fibrosis [15]. 

The use of liver stiffness tools is based on the principle that all tissues have intrinsic mechanical/elastic properties that can be measured by creating a distortion in the tissue and evaluating its response. When the structure of a tissue is modified because of fibrosis deposition, they detect changes in these mechanical properties, the amount of which correlate with the grade of fibrosis. There are two main types of elastography: ultrasound (using ultrasound to detect the velocity of the microdisplacements—shear waves—induced in the tissue) and magnetic resonance elastography [90,91]. In the setting of CH these tools provide unreliable information regarding the grade of fibrosis, since there are other factors that influence the viscoelastic properties of the liver and result in increased liver stiffness, such as the presence of severe hepatic inflammation, biliary obstruction, and congestive HF [85]. Nevertheless, some evidence suggests that liver and spleen stiffness calculated by magnetic resonance elastography may be more accurate [92,93]. Finally, new advances in imaging techniques, such as magnetic resonance imaging with diffusion-weighted imaging, may potentially differentiate fibrosis from congestion but require validation [15].

## 9. Hepatic Hemodynamic Study

Measurement of the HVPG is the gold standard to estimate portal venous pressure. It is widely applied for diagnosing chronic liver disease, assessing the risk of hepatic failure after liver surgery, guiding primary and secondary prophylaxis of variceal bleeding, assessing new therapeutic agents, and providing prognostic information. There is currently no alternative since non-invasive parameters do not estimate portal pressure with similar accuracy [94,95]. 

HVPG represents the difference between the wedged hepatic venous pressure (WHVP) and the free hepatic venous pressure (FHVP). The WHVP is usually measured by occluding the right hepatic vein through the inflation of a balloon, whereas FHVP is measured without occluding it. The occlusion of the vein forms a continuous static column of blood between the catheter and the hepatic sinusoids. Thus, WHVP measures sinusoidal pressure. Due to the scarce connections between sinusoids existing in cirrhosis, pressure cannot be decompressed through the sinusoidal network and, therefore, WHVP reflects portal pressure in this setting. FHVP, on the other hand, is a surrogate for inferior vena cava pressure. In patients with primary liver diseases the HVPG is a strong and independent predictor of outcomes in compensated and decompensated cirrhosis [96,97,98]. The normal HVPG value is between 1 to 5 mmHg. A figure above this range indicates elevated portal pressure and according to their prognostic value, patients with portal hypertension can be classified in two main groups: mild or subclinical (≥6 to 9 mmHg) and clinically significant portal hypertension (≥10 mmHg). The main disadvantages of this technique are its invasiveness and that it requires specific expertise and setting, all of which limit its universal applicability, especially in nonteaching centers [94,95]. 

In patients with CH, the diagnostic and prognostic value of HVPG measurement has not been adequately assessed. In this setting, both FHVP and WHPV are elevated, and the HVPG is within the normal range (Figure 8). Once cardiac cirrhosis is established, the HVPG is expected to increase beyond 6 mmHg (Figure 9) [15]. Hence, HVPG could theoretically provide relevant information about the stage of CH. The few clinical studies that have provided hemodynamic data in this regard have described inconsistent results. For instance, in the study of Myers et al. esophageal varices were seen in some patients despite having a HVPG below 6 mmHg. As previously explained, the high pressures along the entire path of venous return to the right atrium prevent the formation of varices, unless the establishment of cirrhosis creates a pressure gradient between the portal and systemic circulation. In order to explain these discordant results, the same authors argued that it was possible that the varices observed in a few patients represented either false-positive endoscopies or undetected concomitant disease, such as portal vein thrombosis [11]. Moreover, it has not yet been demonstrated that the HVPG correlates with the stage of fibrosis in CH [11,76]. These findings probably respond to several confounders: the inclusion of few patients with advanced fibrosis, the variable distribution of fibrosis throughout the liver, and the absence of a full and reliable characterization of the liver disease. Moreover, in the setting of Fontan-associated liver disease the HVPG can be underestimated due to the frequent presence of vascular fistulas between the hepatic veins themselves or between these veins and the portal branches [12]. As far as its prognostic utility is concerned, no study has evaluated the HVPG for predicting hepatic decompensation events and survival after isolated HT [15]. Despite this, many academic centers, including our own, measure the HVPG to assist in the transplant decision-making process. Finally, it must be reminded that the hepatic vein catheterization also allows performing a transjugular liver biopsy. This technique is safer than the percutaneous biopsy and can be performed, even under anticoagulation or ascites [99]. 

## 10. Prognosis and Treatment

The underlying cardiac disease generally determines prognosis in CH. Liver enzymes (i.e., bilirubin, alkaline phosphatase, gamma-glutamyl transferase, and albumin) and scores such as the MELD and MELD-XI have been associated with prognosis in HF patients [18,21,100,101,102,103]. Based on these findings, both the American College of Cardiology and European Society of Cardiology Heart Failure Guidelines recommend the inclusion of liver function tests in the diagnostic workup of all patients presenting with HF [104,105]. However, it must be pointed out that they predict cardiac or overall mortality, not liver-related mortality. Therefore, they seem to act as indirect markers of the severity of cardiac disease rather than reflecting the effect of liver disease on outcomes. Indeed, the effect of cardiac cirrhosis on overall prognosis has not been clearly established [4]. As far as the prognosis of ACLI is concerned, it is usually poor with an overall hospital mortality of 51% [59] and 1-year survival rate of approximately 25% [5]. The cause of death is usually the underlying condition, as it is an uncommon cause of acute liver failure (only 4.4% of the cases in a study from the Acute Liver Failure Study Group) [106]. 

Management of the underlying cardiac disease is the mainstay of treatment. There is no specific therapy of CH [10]. Concerns about modification of drug dosage have been raised, although there are no solid rules in this regard. This is partially explained by the lack of correlation of available diagnostic tools with the hepatic function [3]. Theoretically more relevant are the detrimental effects that some of the medical therapies used to treat HF may have on the physiopathology of cirrhosis. For instance, vasodilators such as angiotensin-converting-enzyme inhibitors are contraindicated in decompensated cirrhosis and doses of diuretics in HF are often higher than in cirrhosis and may precipitate hepatorenal syndrome [1]. Again, no solid recommendations are available and treatment modifications should be patient specific. 

Finally, in patients with ACLI the management of the underlying diseases remains the only established treatment for ACLI. Although data are limited, some experts recommend using N-acetylcysteine, avoiding excessive vascular filling to minimize passive congestion of the liver, and favoring the use of dobutamine in patients with low cardiac index given its inotropic and vasodilating effects [1,5,8,53].

## 11. Determining Candidacy for Heart Transplantation

Evidence coming from case series and cohort studies using historical controls have shown high morbidity and mortality rates in patients with cirrhosis who undergo nontransplant cardiac surgery, especially in patients with decompensated cirrhosis [107,108,109]. Thus, a MELD score >13 or a CTP score >7 is generally considered a contraindication to cardiac surgery [110]. Scant evidence also shows a higher mortality of patients with cirrhosis undergoing isolated HT [111]. Not surprisingly, for most teams, the diagnosis of cirrhosis is considered a contraindication for HT [112]. In the setting of CH, there is even less data regarding outcomes of isolated HT in patients with established cardiac cirrhosis, but it may also lead to poorer outcomes [113]. The aforementioned limitations of available invasive and non-invasive tests to assess hepatic fibrosis and function make it especially challenging to determine whether a patient with CH is a candidate for isolated HT or may require a CHLT. To make matters worse, there are no official guidelines, evaluation is institution dependent, and the decision is often taken on a case-by-case basis. 

The level of liver involvement that prevents a HT or warrants consideration for CHLT is unclear, with many centers indicating the latter when there is an established cirrhosis on liver biopsy and/or presence of clinically significant portal hypertension. It must be highlighted that cardiac cirrhosis may be reversed after HT. What is still undefined is the subgroup of patients with compensated cirrhosis of any etiology in whom regression to a non-cirrhotic stage is improbable. It has been postulated that, once clinically significant portal hypertension has developed cirrhosis may no longer be reversible, since the thicker fibrous septa seen at this stage are unlikely to regress [114,115]. Based on this premise, some centers use an HVPG value of >12 mm Hg as a cutoff for declining a HT or offering CHLT [15]. This approach also takes into account the increased risk of decompensation and mortality after elective hepatic and extrahepatic surgery in cirrhotic patients with severe portal hypertension [116,117]. Nevertheless, this hemodynamic-based protocol requires validation before its widespread use in clinical practice. Figure 10 shows our protocol for determining our recommendation regarding liver disease in a potential candidate for a HT when CH is suspected. 

The main three indications for CHLT are familial amyloid polyneuropathy (FAP), HF with cardiac cirrhosis (including congenital heart defects that required Fontan procedure), and HF with concomitant noncardiac cirrhosis. In FAP, the liver is transplanted to avoid ongoing damage to the cardiac allograft. A consistent observation regardless of the indication is that cardiac dysfunction is the primary driver of combined organ transplant [118,119,120]. Until recently, FAP remained the most common indication of CHLT in the US. However, CHLT in patients with congenital heart diseases has surpassed non-congenital heart diseases as the leading indication for CHLT in this country for the first three months of 2020 [121]. This disproportionate five-fold increase is due to the growing number of the Fontan-palliated population presenting with decompensated heart failure (there are now more adults than children living with congenital heart diseases in the US), but also to organ allocation policy changes giving them priority status and mandating that, in patients with multiorgan transplantation, the second-required organ has to be allocated to the multi-organ candidate from the same donor [121]. Despite the number of CHLT has increased gradually in the US to approximately 25 cases per year from 2015, CHLT remains a small percentage (<4%) of the total HT worldwide [118]. In Spain, the first CHLT was performed in 1999, and, as of 2019, only 14 CHLT have been performed in four of the 25 institutions harboring a liver transplant program (data provided by the Organización Nacional de Trasplantes). As far as the survival of patients undergoing CHLT is concerned, the overwhelming majority of data from multiple recent studies suggest a similar to improved survival for CHLT compared with isolated HT. Moreover, CHLT seems to have lower rates of cardiac rejection and cardiac allograft vasculopathy compared with isolated HT, indicating an immunologic benefit [121,122,123,124,125]. The underlying mechanism of this immune tolerant property of the liver allograft is not well understood. The leading theory is that it may promote clearance of donor-specific antibodies of the recipients [118,121]. The optimal surgical approach and timing of CHLT remain to be defined, and there are no specific surgical recommendations. In general, there is no major difference in the surgical techniques with regard to the methods of transplantation for each organ. The most commonly described technique is performing the HT on cardiopulmonary bypass, discontinuing bypass, leaving the chest open, and performing the liver transplant with selective use of venovenous bypass [112,118]. Of note, the number of transplant centers with experience in performing this surgery in any national organ transplant system is small, and among these centers, the number of CHLT performed each year varies greatly. In the US, the higher-volume centers perform 10 times the annual rate of lower-volume centers, which, on average, is one every 10 years. This disparity may have some impact in survival as a recent study showed a trend to reduced mortality in CHLT performed in higher-volume centers [121]. 

## 12. Conclusions

Despite the great progress in the knowledge of the pathophysiology, clinical presentation, diagnosis, and treatment of CH, there are still important gaps in all of these areas. Future directions include knowledge of the true burden of CH, identification of validated markers for the presence of liver fibrosis and predicting clinical outcomes, development of uniform criteria for CHLT candidacy, and elucidation of the liver allograft’s immunoprotective mechanisms. This agenda will necessarily require a multidisciplinary approach to overcome the multiple obstacles posed by this complex liver disease.

## Figures and Tables

**Figure 1 ijms-21-09420-f001:**
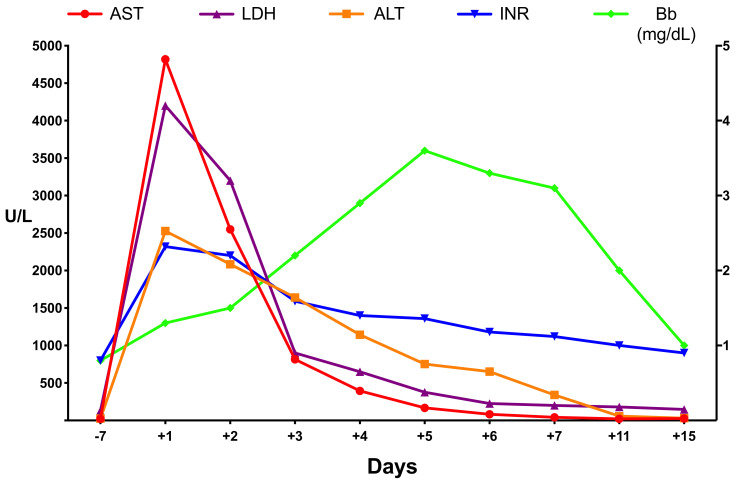
Laboratory parameters during the course of acute cardiogenic liver injury (ACLI) in a patient with respiratory failure due to drug overdose. Abbreviations: AST: aspartate aminotransferase; ALT: alanine aminotransferase; LDH: lactate dehydrogenase; Bb; bilirubin; INR: international normalized ratio.

**Figure 2 ijms-21-09420-f002:**
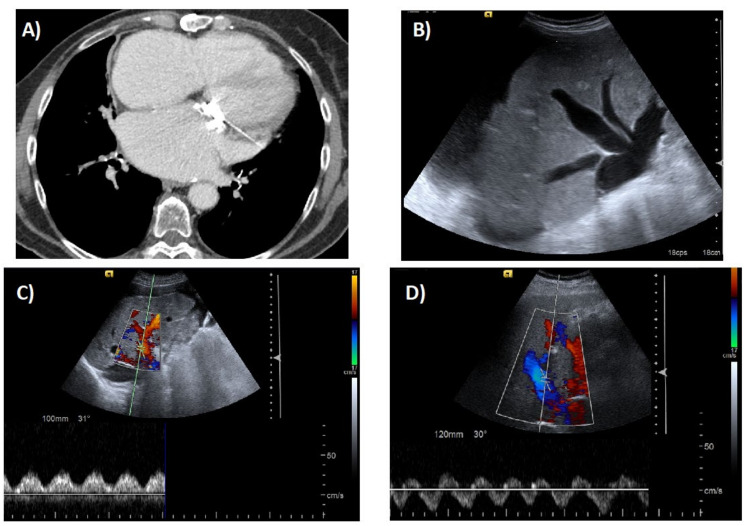
(**A**) Cardiomegaly in a patient with ischemic cardiomyopathy. (**B**) Dilated suprahepatic vein in the same patient. (**C**) Doppler ultrasound in one of the dilated suprahepatic veins. (**D**) Hepatopetal flow in the portal vein highly modulated by the cardiac cycle.

**Figure 3 ijms-21-09420-f003:**
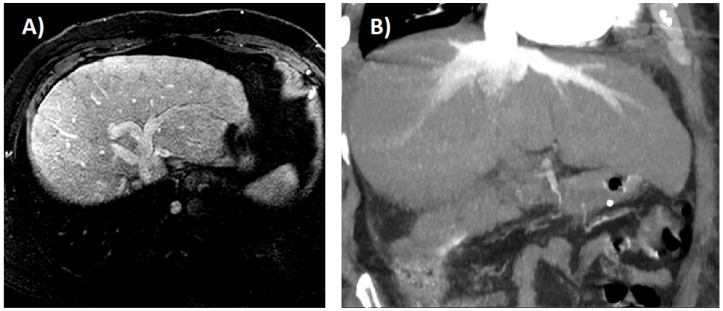
(**A**) Idiopathic membranous inferior vena cava obstruction in a 44-year-old man. Magnetic resonance imaging shows a mildly nodular liver with altered parenchymal perfusion and dilatation of hepatic veins. (**B**) Severe tricuspid regurgitation in a 49-year-old man. Computed tomography scan shows dilatation of hepatic veins and reflux of contrast into the inferior vena cava and hepatic veins.

**Figure 4 ijms-21-09420-f004:**
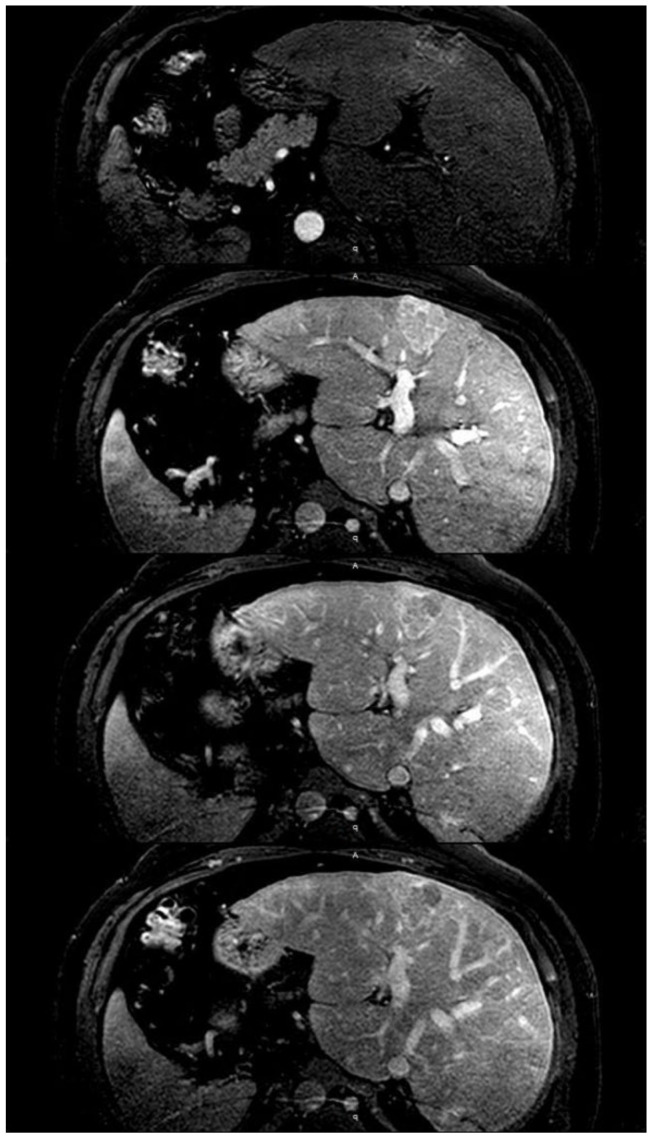
Idiopathic membranous inferior vena cava obstruction in a 44-year-old man. The image shows the dynamic phase of MRI. In addition to the significant hypertrophy of segment I, magnetic resonance imaging shows a mass (3.8 cm × 4.2 cm) that after administration of intravenous contrast presents a heterogeneous enhancement in the arterial phase with washout in the portal phase. Liver biopsy showed histological changes compatible with focal nodular hyperplasia.

**Figure 5 ijms-21-09420-f005:**
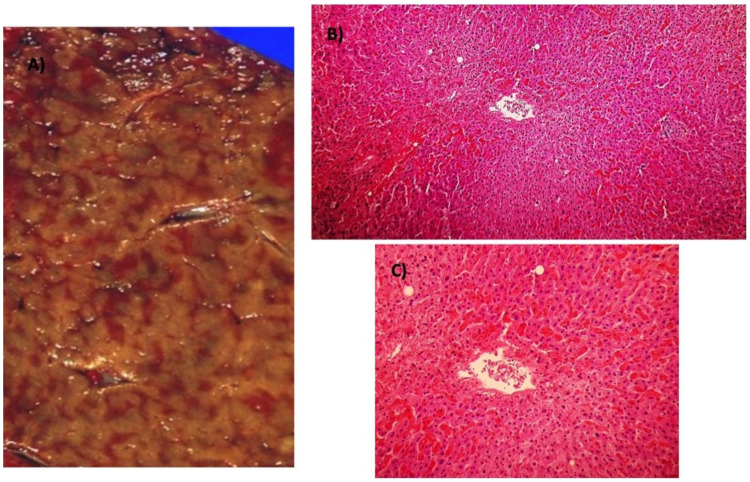
(**A**) Postmortem example of the classical “nutmeg” liver with centrilobular congestion in congestive hepatopathy (CH). (**B**) Centrilobular regions show congestion and extravasation of red blood cells (4× objective). (**C**) Same findings as (**B**), with greater magnification (10× objective).

**Figure 6 ijms-21-09420-f006:**
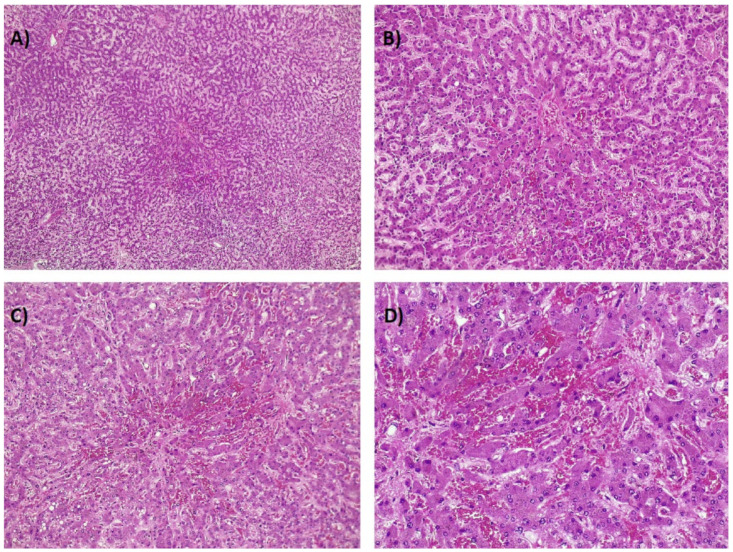
(**A**–**D**). Postmortem example of a patient with sinusoidal obstruction syndrome. As in Figure 5, centrilobular regions show congestion and extravasation of red blood cells (increasing magnification from (**A**) to (**D**): 4×, 10×, 20×, and 40×; hematoxylin-eosin stain).

**Figure 7 ijms-21-09420-f007:**
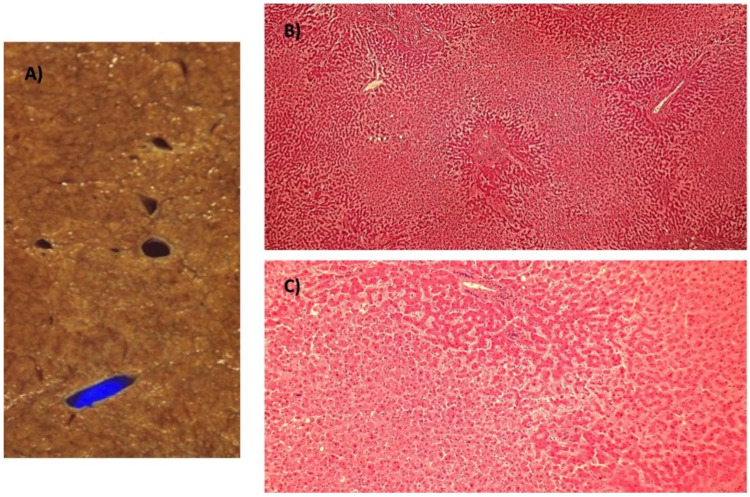
(**A**) Postmortem example of a liver with ischemic zones around centrilobular veins. (**B**) Centrilobular regions show congestion and coagulative necrosis (hematoxylin-eosin stain, 4× objective). (**C**) Same findings as 7B, with greater magnification (10× objective).

**Figure 8 ijms-21-09420-f008:**
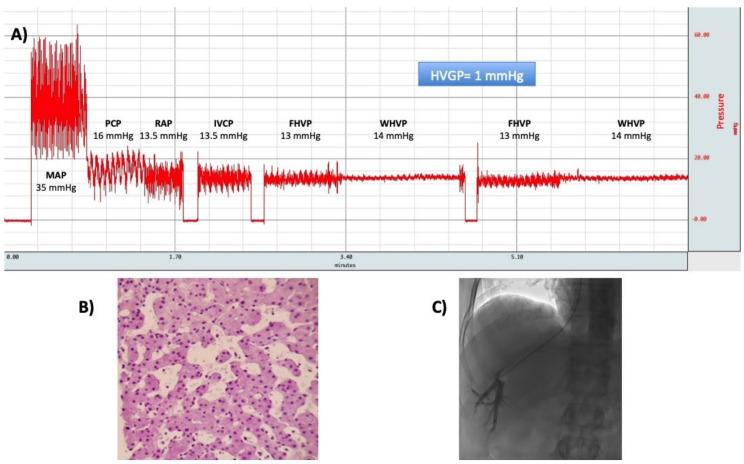
(**A**) A typical hemodynamic tracing of a patient with congestive hepatopathy due to cor pulmonale. The hepatic venous pressure gradient (HVPG) is calculated as the difference between wedged hepatic venous pressure (WHVP), and free hepatic venous pressure (FHVP). Both of them are elevated, but the HVPG is within the normal range. (**B**) Transjugular liver biopsy was performed and showed sinusoidal dilatation without significant fibrosis (hematoxylin-eosin stain, ×20 objective). (**C**) Occlusion of the hepatic vein with the balloon catheter. Abbreviations: MAP: mean pulmonary arterial pressure; PCP: pulmonary capillary pressure; RAP: right atrial pressure; IVCP: inferior vena cava pressure; FHVP: free hepatic venous pressure; WHVP: wedged hepatic venous pressure; HVPG: hepatic venous pressure gradient.

**Figure 9 ijms-21-09420-f009:**
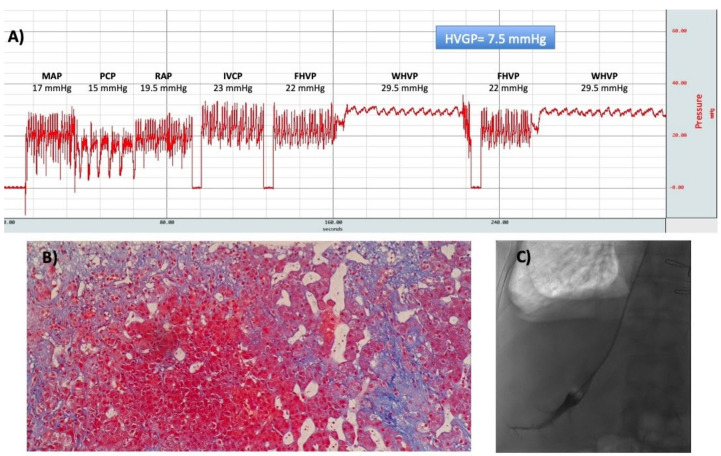
(**A**) A typical hemodynamic tracing of a patient with severe tricuspid regurgitation and concomitant hepatitis C. The HVPG is calculated as the difference between WHVP and FHVP. Both of them are elevated, and the HVPG is slightly elevated. (**B**) Transjugular liver biopsy was performed and showed significant fibrosis forming nodules (Masson stain, ×10 objective). (**C**) Occlusion of the hepatic vein with the balloon catheter. Abbreviations: MAP: mean pulmonary arterial pressure; PCP: pulmonary capillary pressure; RAP: right atrial pressure; IVCP: inferior vena cava pressure; FHVP: free hepatic venous pressure; WHVP: wedged hepatic venous pressure; HVPG: hepatic venous pressure gradient.

**Figure 10 ijms-21-09420-f010:**
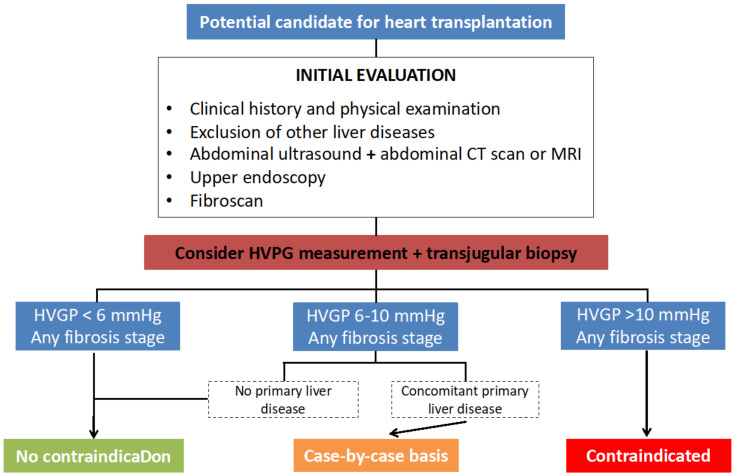
Protocol to determine the recommendation regarding liver disease in a potential candidate for a heart transplant when CH is suspected. We proceed to HVPG measurement and transjugular biopsy in those patients in whom advanced liver disease cannot be ruled out after the initial evaluation (e.g., nodular appearance of the liver). Our recommendation is hemodynamic-dependent, regardless of the fibrosis stage. In cases with a HVPG below 5 mmHg, there is no contraindication to perform an isolated heart transplant, whereas a HVPG > 10 mmHg discards it (no combined heart-liver transplantation has been performed so far in our hospital). In patients with a concomitant primary liver disease and a HVPG between 6–10 mmHg, the decision is patient-specific and relies mainly on the type of disease. If it is treatable (e.g., hepatitis C or B), we recommend proceeding with the heart transplant. The same recommendation is given in the absence of a primary liver disease and a HVPG between 6–10 mmHg. Abbreviations: CT: computed tomography; MRI: magnetic resonance imaging; HVPG: hepatic venous pressure gradient.

**Table 1 ijms-21-09420-t001:** Tests recommended by the American Heart Association for surveillance of liver disease in post-Fontan patients.

	Basic *	In-Depth *	Investigational *
Childhood (every 3–4 years)	CMP Platelet count Serum GGT	PT/INR Serum FibroSure biomarkers Serum α-fetoprotein Abdominal ultrasound Total serum cholesterol	Liver imaging via CT or MRI Liver elastography (ultrasound or MRI) Liver biopsy
Adolescence (every 1–3 years)	CMP Platelet count Serum GGT PT/INR	Serum FibroSure biomarkers Serum α-fetoprotein Abdominal ultrasound Total serum cholesterol Liver imaging via CT or MRI Liver elastography (ultrasound or MRI)	Liver biopsy
Adulthood (every 1–2 years)	CMP Platelet count Serum GGT PT/INR Total serum colesterol Abdominal ultrasound	Serum FibroSure biomarkers Serum α-fetoprotein Liver imaging via CT or MRI Liver elastography (ultrasound or MRI)	Liver biopsy

* Test are stratified as basic (fundamental and rudimentary level of assessment), in-depth (more detailed level of characterization), and investigational (possible or likely of value; however, greater experience and study may be necessary before widespread use can be suggested). Abbreviations: CMP: comprehensive metabolic panel; CT: computed tomography; GGT: γ-glutamyl transferase; INR: international normalized ratio; MRI: magnetic resonance imaging; PT: prothrombin time.

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
