# Peer review of "Congestive Hepatopathy"

_ijms, 2020, doi:10.3390/ijms21249420_

Round 1
Reviewer 1 Report
Fortea et al., have submitted a review article on Congestive Hepatopathy (CH). In this article, they have pathophysiology of the disease in sufficient detail and have discussed clinical diagnosis along with biochemical and histological profiles. They have also discussed imaging, histological, and hemodynamic studies to identify/diagnose CH. Finally, the authors outline current disease management and treatment strategies.
One topic that is left out in the review is the involvement of cardiokines in CH. For example, there is evidence for the presence of atrial natriuretic peptide (ANP) receptors in the liver and the role of ANP in regulation of hepatic metabolism. It would be useful to add a section on "advances/hypothesis" and discuss intra-tissue communication that is generally prevalent in diseases that involve both heart and liver.
Overall, this review is lucidly written, well organized, and covers all major topics related to CH. This work can be accepted in present form but will be improved with the addition of aforementioned topic.
Reviewer 2 Report
Dear Dr.
Editor,
Overall recommendation:
Accept
Final comments:
This paper shows congestive heart disease induced hepatopaty. Their review looks very clear and interesting. Both hepatologists and cardiologists should care this disease, but there is no specialist in this field, and this paper is very helpful to manage chronic heart failure induced liver cirrhosis.
I think this paper is good for publication in this present form.
Kansai Medical University
Katsunori Yoshida
